# Direct medical costs after surgical or nonsurgical treatment for degenerative lumbar spinal disease: A nationwide matched cohort study with a 10-year follow-up

Chi Heon Kim[1,2], Chun Kee Chung[1,2,3]*, Yunhee Choi[4], Juhee Lee[4], Seung Heon Yang[1,2], Chang Hyun Lee[1,2], Sung Bae Park[1,2,5], Kyoung-Tae Kim[6,7], John M. Rhee[8], Moon Soo Park[9]

1 Department of Neurosurgery, Seoul National University Hospital, Seoul, Republic of Korea, 2 Department of Neurosurgery, Seoul National University College of Medicine, Seoul, Republic of Korea, 3 Department of Brain and Cognitive Sciences, Seoul National University, Seoul, Republic of Korea, 4 Division of Medical Statistics, Medical Research Collaborating Center, Seoul National University Hospital, Seoul, Republic of Korea, 5 Department of Neurosurgery, Boramae Medical Center, Seoul National University Boramae Hospital, Seoul, Republic of Korea, 6 Department of Neurosurgery, Kyungpook National University Hospital, Daegu, South Korea, 7 Department of Neurosurgery, School of Medicine, Kyungpook National University, Daegu, South Korea, 8 Department of Orthopaedic Surgery, Emory University School of Medicine, Atlanta, Georgia, United States of America, 9 Department of Orthopedics, Hallym University Dongtan Sacred Heart Hospital, Gyeonggi, Republic of Korea

* chungc@snu.ac.kr

**Data Availability Statement:** The National Health Insurance System-National Sample Cohort (NHIS-NSC) was utilized for this study after approval by

## Abstract

### Objective

The demand for treating degenerative lumbar spinal disease has been increasing, leading to increased utilization of medical resources. Thus, we need to understand how the budget of insurance is currently used. The objective of the present study is to overview the utilization of the National Health Insurance Service (NHIS) by providing the direct insured cost between patients receiving surgery and patients receiving nonsurgical treatment for degenerative lumbar disease.

### Methods

The NHIS-National Sample Cohort was utilized to select patients with lumbar disc herniation, spinal stenosis, spondylolisthesis or spondylolysis. A matched cohort study design was used to show direct medical costs of surgery (n = 2,698) and nonsurgical (n = 2,698) cohorts. Non-surgical treatment included medication, physiotherapy, injection, and chiropractic. The monthly costs of the surgery cohort and nonsurgical cohort were presented at initial treatment, posttreatment 1, 3, 6, 9, and 12 months and yearly thereafter for 10 years.

### Results

The characteristics and matching factors were well-balanced between the matched cohorts. Overall, surgery cohort spent $50.84/patient/month, while the nonsurgical cohort spent

health insurance review and assessment service (HIRA). Individual data linkage of population was made internally in the Big Data Steering Department of the National Health Insurance Service. The authors of the study were approved to use customized tables via virtual terminal connected to personal computer after review of study proposal by HIRA for less than 6 months. By law, sharing a raw data or copying the data including photo copy is strictly banned. Therefore, we could not upload raw data in Plos One submission system. I added dataset as Supporting information in revision, but raw data could not be included. Any research who had interest in this study can request the use of NHIS-NSC by following the procedures outlined at homepage of HIRA (https://opendata.hira.or.kr/op/opc/selectPatDataApIInfoView.do). There is a cost for the use of virtual terminal.

**Funding:** This work was financially supported by the New Faculty Startup Fund ($ 35,000) from Seoul National University (CHK), Seoul 03080, Republic of Korea. A grant ($1,300,000) from the Korea Health Technology Research & Development Project supported this study through the Korea Health Industry Development Institute (KHIDI) funded by the Ministry of Health & Welfare, Republic of Korea (HC15C1320) (CKC). The funders had no role in study design, data collection and analysis, decision to publish, or preparation of the manuscript. No authors in this study received salary or materials by funders.

**Competing interests:** The authors have declared that no competing interests exist.

**Abbreviations:** HIRA, Health Insurance Review & Assessment Service; LDH, lumbar disc herniation; NHIC, National Health Insurance Corporation; NHID, National Health Insurance database; NHIS-NSC, National Health Insurance Service-National Sample Cohort; ROK, Republic of Korea.

$29.34/patient/month (p<0.01). Initially, surgery treatment led to more charge to NHIS ($2,762) than nonsurgical treatment ($180.4) (p<0.01). Compared with the non-surgical cohort, the surgery cohort charged $33/month more for the first 3 months, charged less at 12 months, and charged approximately the same over the course of 10 years.

## Conclusion

Surgical treatment initially led to more government reimbursement than nonsurgical treatment, but the charges during follow-up period were not different. The results of the present study should be interpreted in light of the costs of medical services, indirect costs, societal cost, quality of life and societal willingness to pay in each country. The monetary figures are implied to be actual economic costs but those in the reimbursement system instead reflect reimbursement charges from the government.

## Introduction

The demand for degenerative spinal disease treatments has been increasing and has led to the increased utilization of medical resources, including surgical treatments and nonsurgical treatments, such as exercise, physical therapy, medication and other interventions [1–3]. Many countries have been trying to provide optimal medical services to their residents, but there is also an issue of how to utilize limited resources, especially in regard to budgets [4–7]. A National Health Insurance System (NHIS) is one tool that the government can use to take responsibility in providing medical services to the general population while balancing the efficient use of resources [8, 9]. Ideally, from the perspective of society, we need to understand how the budget is currently being used [5]. In the Republic of Korea (ROK), all citizens have been beneficiaries of the national health insurance system (NHIS) for more than 20 years, and the NHIS covers both Western and Oriental medicine [8–10]. Because the NHIS follows a fee-for-service payment system, all nationwide inpatient and outpatient data on diseases and services (i.e., procedures and operations) are coded and registered in the National Health Insurance Corporation (NHIC) database and the Health Insurance Review & Assessment Service (HIRA) database [1, 8–12]. By using the database, the National Health Insurance Service-National Sample Cohort (NHIS-NSC) was identified in 2017 for analysis while maintaining representativeness and protecting personal information [12]. The objective of the present study was to provide an overview of the utilization of the NHIS for degenerative lumbar spinal disease by reporting the direct medical costs of patients who underwent surgery and of patients who did not undergo surgery.

## Materials and methods

### Study design

A matched cohort study design was used for the present study. Initially, patients were selected from the NHIS-NSC on the basis of diagnosis and surgery codes. Thereafter, the following factors were considered for matching: age, sex, diagnosis, osteoporosis without fractures and diabetes mellitus, and the total charge incurred over the previous 3 years (which was assumed to be a surrogate for disease severity) [9, 10, 13, 14]. Osteoporosis and diabetes mellitus were specifically included for matching, because those factors were associated with outcomes in previous studies [9, 10, 13, 14]. Because clam data included many disease codes without

information of severity, the other codes were not included in matching process. To match with limited factors, the categorical factors were matched exactly. The continuous variables were matched with a difference of no more than 3 years for age and 10% of standardized difference of total direct medical cost incurred in patients with surgery over the previous 3 years by lumbar spinal degenerative disease. The matching within 10% of the standard deviation of total direct medical cost was to make the standardized difference between the two groups less than 10%, which is considered well balanced between two groups. This study used NHIS-NSC data (NHIS-3017-2-494) generated by the National Health Insurance Service (NHIS). The requirement for informed consent was waived because the data were deidentified, and the Institutional Review Board approved the review and analysis of the data (No. 2010-076-1164). All methods were carried out in accordance with relevant guidelines and regulations.

## Data source

The National Health Insurance database (NHID) was developed to record personal information, demographics and medical treatment data for all Korean citizens [12, 13, 15]. The disease codes were standardized based on the 10th version of the International Classification of Diseases (ICD-10), and the procedure codes were standardized to the claim medical fees [9, 10, 14]. The NHIC set guidelines encouraging nonsurgical treatment to be performed for at least 6 weeks before surgery for patients with lumbar disc herniation (LDH) and 3 months for patients with lumbar spinal stenosis with or without spondylolisthesis [8, 9]. Nearly all hospitals providing Western medicine and clinics providing Oriental medicine must follow the guidelines to obtain reimbursement. The detailed surgical and nonsurgical management were determined by the attending physicians [9, 10, 14]. The NHIS-NSC is a representative sample cohort, and 1,000,000 people (2.1% of the total Korean population) were randomly selected from a total population of 48,438,292 in 2006 (https://nhiss.nhis.or.kr/bd/ab/bdaba021eng.do) [12]. Systematic stratified random sampling with proportional allocation within each stratum was conducted [12]. The strata included those for sex, age, location, and health insurance type (insured employees, insured self-employed individuals or medical aid beneficiaries) [12]. The resident registration number was replaced with a newly assigned eight-digit personal ID, which enables longitudinal follow-ups to be performed for all people until 2015 [12]. During the follow-up period, the cohort was updated annually; a representative sample of newborns was included, and the size of the cohort was maintained [12]. The same cohort included any claims from hospitals, pharmacies and Oriental medicine clinics. The records for each person in the NHIS-NSC can be traced back to 2002.

## Study group

The present study included patients diagnosed with LDH, lumbar spinal stenosis without spondylolisthesis, lumbar spinal stenosis with spondylolisthesis, and spondylolysis, which followed the hierarchical coding algorithm for the diagnosis proposed by Martin et al. [16] The surgery cohort was selected in the following way: initially, patients who underwent surgery between 2006 and 2008 were searched with specific procedure codes (open discectomy N1493; laminectomy, N1499 and N2499; endoscopic lumbar discectomy, N1494; spinal fusion, N0466, N1466, N0469, N2470, N1460 or N1469) and disease codes as a primary or secondary diagnosis (LDH, M51, M472; lumbar spinal stenosis without spondylolisthesis, M4800, M4805-8; lumbar spinal stenosis with spondylolisthesis, M431, M4315-9; spondylolysis, M430, M4306-9), and 4,577 patients were selected. Among them, patients with the following conditions were excluded, and 3,881 patients remained in the surgery cohort: 1) a history of spinal surgery within the past 3 years, 2) a history of using medical services with disease codes for

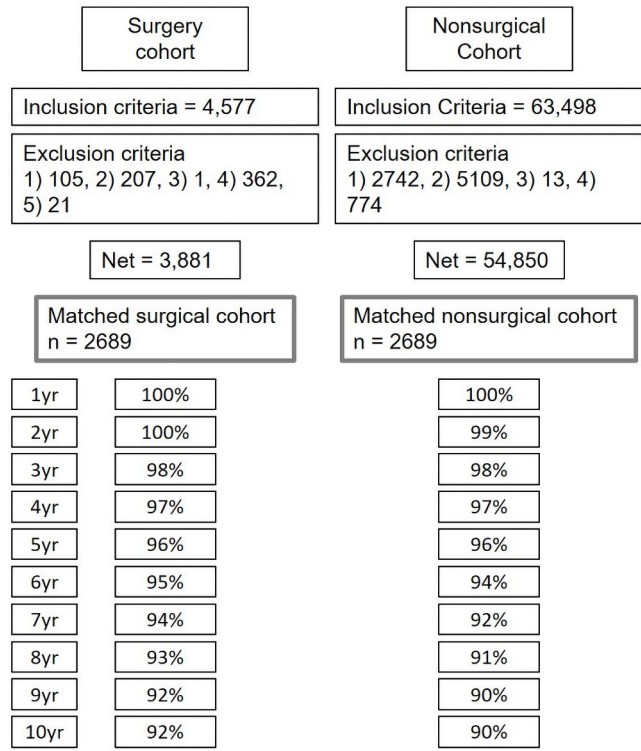

**Fig 1. Flow diagram of patients.** The number of patients were described in the box. The follow-up rates were marked along with time points. The detailed inclusion and exclusion criteria are described in the manuscript.

spinal fracture, pathological fracture, spinal infection, any kind of malignancy, or inflammatory joint disease within the past 1 year, 3) a concomitant rare disease such as a metabolic disease, blood disease, or congenital anomaly, 4) admission via the emergency room, and 5) an age less than 18 years (Fig 1). During the last 3 years, the standard deviation of the direct medical cost (charge to NHIS) in 3,881 patients was $730, and a difference of no more than $73 was considered a similar severity of disease in the matching process. The nonsurgical cohort was selected in the following way (Fig 1): the patients who visited clinics more than 3 times/year or were admitted to the hospital more than 2 days/year with the same diagnosis as the surgery cohort were searched, and 63,498 patients were selected. Among them, patients with the following conditions were excluded, and 54,850 patients remained in the nonsurgical cohort: 1) a history of spinal surgery within the past 3 years, 2) a history of using medical services with disease codes of spinal fracture, pathological fracture, spinal infection, any kinds of malignancy, or inflammatory joint disease within the past 1 year, 3) a concomitant rare disease such as a metabolic disease, blood disease, or congenital anomaly, and 4) an age less than 18 years. After matching, 2,698 patients remained in each cohort, and each patient was individually followed for 10 years with their unique ID (Fig 1). Data on the direct medical cost used for both Western and Oriental medicine were retrieved from the database. The costs here do not account for societal costs such as missed work or quality of life. Treatment failure was defined as any kind of lumbar spinal surgery at any lumbar level being performed after the initiation of either surgical or nonsurgical treatment [8, 9, 11, 14]. Because the exact lumbar level was not recorded in the registry, treatment failure after surgery was included at both the index surgery (reoperation) and the other lumbar levels [8, 9, 11, 13, 14].

## National Health Insurance System in Republic of Korea [14]

The ROK adopted a government-controlled NHIS, and it is funded by the taxpaying citizenries. The NHIS is a service provider, and the Health Insurance Review & Assessment Service (HIRA, https://www.hira.or.kr/eng/main.do) controls approvals of reimbursement. In Korea, the hospital type is defined by law [14]. General hospitals have at least 7 departments with more than 99 beds, including internal medicine, general surgery, obstetrics and gynecology, pediatrics, diagnostic radiology, anesthesiology, pathology, and laboratory medicine, with at least 1 board-certified doctor in each department [9]. Tertiary-referral hospitals are designated from among the general hospitals by the government. A tertiary-referral hospital should have at least 20 departments and should include the basic requirements of a general hospital along with having a residency training program, at least 5 operating rooms, and a variety of imaging/ diagnostic tools used for computed tomography, magnetic resonance imaging, electromyography, angiography, gamma camera radiography, and Holter cardiac monitoring. In addition, the proportion of patients with difficult diseases (as designated by the Minister of Health and Welfare) should be more than 12% of the total number of annual inpatients [9, 14]. The final type, hospitals, is defined as a hospital lacking any of the essential departments or having between 30 and 99 beds. Private clinics have fewer than 30 beds [9]. The NHIS allows people free to choose medical service providers. The deduction rate is 50% for outpatient clinics and 20% for admission (https://hineca.kr/1913). A "fee for service" (FFS) is the traditional reimbursement system, and hospitals request reimbursement after outpatient clinics or discharge of patients. The fee is set by the relative value of the service, which considers the cost of infrastructure, instruments, supplies and services by medical staff [17, 18]. However, the fee may underestimate an actual cost of hospital to provide medical service. Therefore, the imbursement by NHIS does not represent actual cost of hospital but represents a utilization of NHIS. The reimbursement was approved by the review board in HIRA. For reimbursement, the disease codes were standardized based on the international classification of diseases, 10th version (ICD-10). The codes of medical services are standardized by the NHIC and HIRA to file claims for medical fees to the NHIC. All types of hospitals should follow the standardized codes of diseases and procedures for reimbursement by law. The total insured direct costs (which are covered by the NHIS during admission and clinical visits) are recorded in the HIRA database, but the uninsured direct costs, such as using anti-adhesive agents, hemostatic agents and artificial bone material, are not recorded and are known to comprise 35% of the direct medical cost [19].

## Statistical analysis

The average monthly cost was compared between the surgery cohort and the nonsurgical cohort. For the initial cost during 1 month in nonsurgical cohort, only costs of patients who underwent intervention were included because of high cost of nonsurgical intervention (Fig 1). The cost of all patients in nonsurgical cohort were included at the following times: 3, 6, 9, and 12 months after treatment and yearly thereafter. The initial cost for surgery was defined as the cost incurred during admission for surgery. The costs are represented as the mean (standard deviation, SD), and the values were compared between groups with paired t-tests. The reoperation rate (%) was compared between groups using the McNemar test. The Bonferroni method was applied to control the level of type I error for multiple tests. The patient characteristics were summarized as means (SD) for continuous variables and as frequencies (proportions, %) for categorical variables. The surgical methods included open discectomy, endoscopic discectomy, discectomy and fusion and laminectomy without discectomy for LDH. For the other diagnoses, the surgical methods included decompression only and

**Table 1. Characteristics of the cohorts.**

| Cohorts | Surgery | Nonsurgical |
|---|---|---|
| | n = 2,698 | n = 2,698 |
| **Age, mean (standard deviation)** | 54.85 (14.19) | 54.98 (14.19) |
| **Sex, n (%)** | | |
| Male | 1,162 (43.07) | 1,162 (43.07) |
| Female | 1,536 (56.93) | 1,536 (56.93) |
| **Diagnosis** | | |
| **Disc herniation** | 1,229 (45.55) | 1,229 (45.55) |
| **Lumbar spinal stenosis without spondylolisthesis** | 1,059 (39.25) | 1,059 (39.25) |
| **Lumbar spinal stenosis with spondylolisthesis** | 360 (13.34) | 360 (13.34) |
| **Spondylolysis** | 50 (1.85) | 50 (1.85) |
| **Cost ($) during past 3 years***  | 366.67 (453.58) | 366.47 (453.99) |
| **Osteoporosis without fracture, n (%)** | 763 (28.28) | 763 (28.28) |
| **Diabetes mellitus, n (%)** | 406 (15.05) | 406 (15.05) |

*$1 = 1,150 Korean won

decompression with fusion. Conservative treatments included the following: injection procedure [KK061-, N1495-]) and medication/physiotherapy (prescribed medicine, therapeutic exercise [MM101-] and chiropractic [51040]). All analyses were conducted using SAS, version 9.4 (SAS Institute, Cary, NC), and P < 0.05 (two-tailed) indicated statistical significance.

## Results

The characteristics and matched factors were well balanced between the matched cohorts (Table 1). The median age was 57 years, and females comprised 56% of each cohort. LDH was the most common diagnosis, affecting 45.5% of the cohorts, followed by spinal stenosis without spondylolisthesis (39.3%), spinal stenosis with spondylolisthesis (13.3%) and spondylolysis (2%).

Each patient spent $190 over the past 3 years. Osteoporosis was present in 28% of patients, and diabetes mellitus was present in 15% of patients in each cohort. The surgical methods and nonsurgical methods are described in Table 2. Each patient underwent only one surgical treatment but underwent various nonsurgical treatments. Open discectomy was the preferred surgical method for LDH. Fusion surgery was performed in 12% of spinal stenosis cases without spondylolisthesis and was performed in 35–39% of spondylolisthesis or spondylolysis cases. Nonsurgical intervention was performed in 51% of LDH cases and in 70% of spinal stenosis cases with or without spondylolisthesis (Table 2).

The average monthly direct medical cost in the surgery cohort was $50.84/patient/month, while the direct medical cost in the nonsurgical cohort was $29.34/patient/month (p < 0.01) (Table 3). The average cost of surgery was also significantly higher than that of the intervention ($39.89) (p < 0.01) (Table 3).

Initially, surgery cost more ($2,761.5) than the nonsurgical cohort did ($180.4) (p < 0.01) (Table 4). Compared with the nonsurgical cohort, the surgery cohort paid more by $33/month for the initial 3 months, paid less at 12 months by $18.6/month, and paid a similar amount thereafter for 10 years (Fig 2 and Table 4).

Treatment failure occurred in 15% of the surgery cohort and 9% of the nonsurgical cohort (p < 0.01) (Table 5). When the diagnoses were considered separately, the patients with LDH

**Table 2. Details of the treatments.**

| Diagnosis | Treatment | Surgery cohort N = 2,698 | Nonsurgical cohort N = 2,698 |
|---|---|---|---|
| **Lumbar disc herniation** | Surgery* | **n = 1,229** | **n = 1,229** |
| | Open discectomy | 1,002 (81.53) | |
| | Endoscopic discectomy | 125 (10.17) | |
| | Laminectomy | 53 (4.31) | |
| | Discectomy and fusion | 49 (3.99) | |
| | Nonsurgical treatment† | | |
| | Intervention | 582 (47.36) | 629 (51.18) |
| | Medication/Physiotherapy | 1,213 (98.70) | 1,229 (100.00) |
| **Lumbar spinal stenosis without spondylolisthesis** | Surgery | **n = 1,059** | **n = 1,059** |
| | Decompression | 923 (87.16) | |
| | Decompression and fusion | 136 (12.84) | |
| | Nonsurgical treatment | | |
| | Intervention | 647 (61.10) | 745 (70.35) |
| | Medication/Physiotherapy | 1,052 (99.3) | 1,059 (100.00) |
| **Lumbar spinal stenosis with spondylolisthesis** | Surgery | **n = 360** | **n = 360** |
| | Decompression | 233 (64.72) | |
| | Decompression and fusion | 127 (35.28) | |
| | Nonsurgical treatment | | |
| | Intervention | 220 (61.11) | 259 (71.94) |
| | Medication/Physiotherapy | 360 (100) | 360 (100.00) |
| **Spondylolysis** | Surgery | **n = 50** | **n = 50** |
| | Decompression | 31 (62) | |
| | Decompression and fusion | 19 (39) | |
| | Nonsurgical treatment | | |
| | Intervention | 24 (48.00) | 28 (56) |
| | Medication/Physiotherapy | 50 (100) | 50 (100.00) |

*Each patient underwent only one surgical method.

†Each patient underwent either or both treatments.

and spinal stenosis without spondylolisthesis showed a higher failure rate after surgery than the nonsurgical cohort did, but the patients with spinal stenosis with spondylolisthesis did not have different failure rates, while patients with spondylolysis had lower failure rates.

## Discussion

The objective of the present study was to provide an overview of the medical costs after treatment for degenerative lumbar conditions of the surgery and nonsurgical treatment cohorts. The present study showed that the monthly cost was higher in the surgery cohort than in the nonsurgical cohort. When the costs were broken down according to time periods, surgery was initially costlier than nonsurgical treatment. However, the maintenance cost of the surgery cohort was not higher than that of the intervention cohort over 10 years. The results of the present study need to be interpreted with consideration of the cost of medical services, indirect costs, societal cost such as missed work or patient quality of life, societal willingness to pay (WTP), the guidelines of insurance institutes, and the limitations of the analysis in terms of its use of secondary data, and these issues will be discussed [20].

**Table 3. The monthly cost for each treatment.**

| | | Surgery cohort (N = 2,698) | Nonsurgical cohort (N = 2,698) | | P-value* | P-value† |
|---|---|---|---|---|---|---|
| | | | Intervention (n = 1,661) | Nonsurgical cohort (n = 2,698) | | |
| Lumbar disc herniation | Number | 1229 | 629 | 1229 | | |
| | Cost ($) | 36.29 (45)‡ | 26.53 (34.92) | 18.33 (38.08) | <.01 | <.01 |
| Spinal stenosis without spondylolisthesis | Number | 1059 | 745 | 1059 | | |
| | Cost ($) | 61.46 (85.96) | 47.13 (68.28) | 38.53 (61.94) | <.01 | <.01 |
| Spinal stenosis with spondylolisthesis | Number | 360 | 259 | 360 | | |
| | Cost ($) | 68.28 (55.98) | 51.9 (89.53) | 40.78 (78.47) | <.01 | <.01 |
| Spondylolysis | Number | 50 | 28 | 50 | | |
| | Cost ($) | 57.61 (41.48) | 36.3 (42.56) | 22.8 (35.43) | <.01 | <.01 |
| Total | Number | 2,698 | 1,661 | 2,698 | | |
| | Cost ($) | 50.84 (66.72) | 39.89 (62.77) | 29.34 (55.81) | <.01 | <.01 |

* Surgery cohort vs. Nonsurgical cohort.

† Surgery vs. Intervention.

‡ Mean (standard deviation).

## Surgery vs nonsurgical treatment for spinal degenerative disease

Whether surgical or conservative treatments for degenerative spinal disease should be administered has been a controversial issue [16, 21–26]. This issue has been addressed by randomized controlled trials (RCTs) and cost-effectiveness analyses [5, 16, 21, 22, 24–29]. Weinstein JN et al. organized a randomized controlled trial (Spine Patient Outcomes Research Trial, SPORT) to assess the clinical outcomes of surgical and nonsurgical treatments [25, 26]. The SPORT study reported more favorable clinical outcomes after surgery than after nonsurgical

**Table 4. The monthly cost after the initiation of treatment during each period (units, US dollars).**

| Time | Surgery cohort, mean (SD) | Nonsurgical cohort, mean (SD) | Difference§ |
|---|---|---|---|
| 0 (mo) | 2,761.5 (1,952.1) | 180.4 (460)† | 2,581.1* |
| 3 (mo) | 89.4 (265.5) | 56.4 (162.7)‡ | 33* |
| 6 (mo) | 43.3 (229) | 50.6 (196.9) | -7.3 |
| 9 (mo) | 28.4 (114.3) | 39.1 (148.9) | -10.7 |
| 12 (mo) | 25.1 (114.7) | 43.8 (170) | -18.6* |
| 2 (yr) | 31.2 (108.2) | 38.4 (101.2) | -7.1 |
| 3 (yr) | 26.3 (120.1) | 31.2 (102.8) | -4.9 |
| 4 (yr) | 25.6 (101.2) | 32.3 (98.7) | -6.7 |
| 5 (yr) | 25 (112.5) | 28 (94.1) | -3 |
| 6 (yr) | 31.5 (133.1) | 28.9 (105) | 2.6 |
| 7 (yr) | 33.5 (149.4) | 26.1 (112.3) | 7.4 |
| 8 (yr) | 28.3 (118.8) | 19.9 (103.1) | 8.4 |
| 9 (yr) | 14.2 (84.5) | 10.7 (95.4) | 3.5 |
| 10 (yr) | 3.2 (22.8) | 2.9 (46.7) | 0.3 |

* Statistically significant with a p-value of 0.0036 with Bonferroni correction (= 0.05/14).

† Patients with nonsurgical intervention.

‡ All patients in nonsurgical cohort.

§ Difference between surgery cohort and nonsurgical cohort.

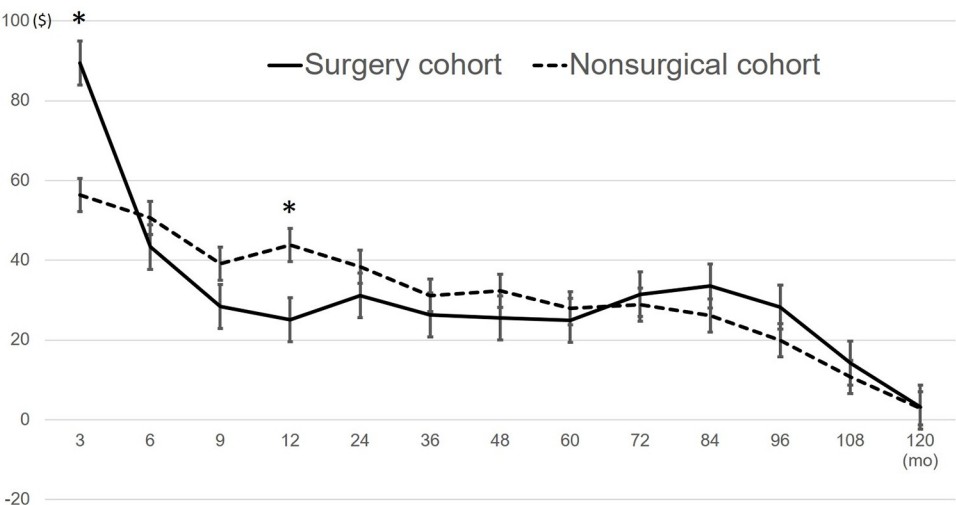

**Fig 2. Monthly cost after treatment.** The average monthly costs and standard errors are plotted in the graph. Asterisk (*) implies statistically significant difference between surgery cohort and nonsurgical cohort.

interventions in patients with spinal stenosis with and without spondylolisthesis [22, 30]. Daffiner S et al. analyzed the costs for 30,709 patients with LDH who received nonsurgical care before ultimately undergoing surgery by using a health insurance database [20]. During the preoperative 3 months, $105,799,925 was spent for nonsurgical treatment ($3,445/patient), and the nonsurgical intervention accounted for 32% of the cost. Considering that the cost of surgery was $7,841, the total cost could have been reduced by early surgery [20]. The costs and clinical outcomes were considered together in a cost-effectiveness analysis [23]. Tosteson AN et al. summarized the cost-effectiveness of surgery: surgery improved quality of life to a greater extent than nonsurgical treatment, and the cost per QALY (quality-adjusted life year) was $20,600 (95% CI: $4,539, $33,088) for LDH, $59,400 (95% CI: $37,059, $125,162) for spinal stenosis without spondylolisthesis, and $64,300 (95% CI: $32,864, $83,117) for spinal stenosis with spondylolisthesis at 4 years [21]. Based on these results, early surgery is recommended as a cost-effective option [20]. Those estimates were based on medical costs, and they would be higher if indirect costs, 75% of which can be attributed to the costs for low back pain, had been considered simultaneously [31]. However, the recommendation was made on the basis of the high costs of nonsurgical medical services. The present results showed that the cost of surgery

**Table 5. Failure of treatment.**

| Cohort | Surgery | | | Nonsurgical | | | P-value |
|---|---|---|---|---|---|---|---|
| | Patients | Failure* | Rate (%) | Patients | Failure | Rate (%) | |
| **Lumbar disc herniation** | 1,229 | 195 | 15.87 | 1,229 | 86 | 7.00 | < 0.01 |
| **Spinal stenosis without Spondylolisthesis** | 1,059 | 171 | 16.15 | 1,059 | 106 | 10.01 | < 0.01 |
| **Spinal stenosis with spondylolisthesis** | 360 | 42 | 11.67 | 360 | 48 | 13.33 | > 0.05 |
| **Spondylolysis** | 50 | 2 | 4.00 | 50 | 5 | 10.00 | > 0.05 |
| **total** | 2,698 | 410 | 15.20 | 2,698 | 245 | 9.08 | < 0.01 |

*Failure after surgery included both surgery at the index level (reoperation) and the other lumbar levels, because the exact lumbar spinal level was not recorded in the registry data.

was $2761.50 and that the cost of nonsurgical medical services, including injection procedures, was $180.40/patient/month. Therefore, according to the NHIS results, conservative treatments may be cost-effective in the ROK. However, the issues of societal cost and indirect cost were not addressed in this study. A fast recovery after surgery, reducing indirect medical costs by shortening the period of job loss, reducing direct medical costs by reducing the cost of nonsurgical management and improving quality of life would offset some of the early costs of surgery [20, 32, 33]. One study from the Sweden health data registry showed that indirect costs comprised 57% of the total cost per patient [34]. Therefore, reducing indirect costs by shortening sick leave from jobs and improving quality of life would effectively reduce total medical costs, and surgery may be a cost-effective option from a long-term economic perspective [32]. Various factors such as the cost of medical services, indirect medical costs and societal WTP vary according to the medical system and economy of each society should be considered together to interpret those results [6, 34–39].

## Use of registered health care data for better management

Although RCTs that focus on clinical outcomes can provide robust recommendations for spinal clinical practice [22, 23, 25, 26], cost-effectiveness is important to physicians, patients and health insurance institutes [7, 23, 36]. These issues are quite serious for spinal disease, considering the recent increase in the number of cases, high cost of surgery/intervention and maintenance cost after treatment [3, 36]. Because the coverage policy of insurance influences the practice pattern, safety and cost of spinal surgery, the stakeholders of insurance institutions should know the current utilization of health care services [5]. Although those issues have been analyzed by using medical records, the costs varied among societies to obtain a generalized picture of the utilization of medical services [21, 24, 40]. This issue could be addressed by using registered data to reveal the big picture [16, 21–24, 26, 41–45]. Recently, claim data have not only been used for accounting but also been used to improve the management of spinal disease [4, 5, 14, 20, 42, 43, 46]. National population-based data can be utilized to accomplish goals, such as analyzing the change in the number of surgical cases, average cost for treatment, use of resources and factors influencing cost [28]. For example, Martin et al. used claim data to indicate a potential overuse of spinal fusion surgery amid a similar reoperation rate and increased surgical cost [29]. The issues of an increased cancer risk and reduced reoperation rate after using bone morphogenic protein in lumbar fusion surgery were addressed by using claims data [27]. Although the analysis of claim data is a retrospective type of analysis, this type of analysis leads to low selection bias by encompassing a large population over a large area and, consequently, providing a robust result [27, 29]. The current study utilized NHIS-NSC data, which represent the entire population in the ROK, to obtain a comprehensive nationwide overview of health care utilization and to provide a practical estimate. The results can be referenced in the future to make budget plans and modify insurance services.

## Strengths and limitations

To the best of our knowledge, the present study is the first to compare direct medical costs between surgery and nonsurgical treatment under NHIS over a long-term follow-up period [12, 13]. A direct medical costs for both Western and Oriental medicine are considered and these results may be helpful in countries with the similar medical system. However, the present study has several limitations. As mentioned, the claim data lack detailed clinical and radiological information, which limits the direct application of the results to clinical practice [5, 28, 29]. Common clinical indicators such as quality of life or disability index would be helpful to better compare surgery and nonsurgical management. Second, the current results cannot be

generalized because the thresholds and indications of surgery and the outcomes of treatments are not uniform across doctors and countries, and the NHIS influences the indications by enforcing reimbursement restrictions [5, 47]. Moreover, claims data do not assess the over/ underuse of treatments by physicians or over/underutilization of insurance services by patients. Third, the costs here do not account for uninsured costs or societal costs such as missed work or quality of life. [13, 48]. Medical fees for services that are not covered by insurance represent approximately 35% of medical costs in the ROK, and the actual direct costs can be estimated as the total cost*(100/65) [19]. Societal costs could not be assessed by a secondary data that lacks clinical information, employment and functional status. In addition, as with most cost-effectiveness studies, specific dollar values might not be equivalent among studies, and the WTP considering the benefits and risks of each surgical method in different countries may lead to different interpretations [35, 43]. Fourth, only age, sex, diagnosis, osteoporosis, diabetes and previous direct medical costs were considered for the matching process, and other important comorbidities, such as smoking, hypertension and congestive heart failure, and socioeconomic factors such as income and employment status, were not considered. Although the result seemed to provide a general overview of the direct medical costs, the current results posed a risk of bias due to potential incomplete matching. Lastly, but not least, this paper has a limitation that true determination of cost requires a thorough accounting including time-driven activity-based costing, which is far outside the scope of this paper. The present monetary figures were a charge to government for a reimbursement, which does not reflect the actual economic cost. Regardless of these limitations, the present study provides an overview of how the NHIS is utilized for lumbar degenerative spinal disease on a nationwide scale, and this information may be helpful for physicians, patients and policy makers to improve the NHIS.

## Conclusion

Surgical treatment initially led to more government reimbursement than nonsurgical treatment, but the charges during follow-up period were not different. The results of the present study should be interpreted in light of the costs of medical services, indirect costs, societal cost, quality of life and societal WTP in each country. The monetary figures are implied to be actual economic costs but those in the reimbursement system instead reflect reimbursement charges from the government.

## Supporting information

**S1 Data. The National Health Insurance System-National Sample Cohort (NHIS-NSC) was utilized for this study after approval by health insurance review and assessment service (HIRA).** Individual data linkage of population was made internally in the Big Data Steering Department of the National Health Insurance Service. The authors of the study were approved to use customized tables via virtual terminal connected to personal computer after review of study proposal by HIRA for less than 6 months. By law, sharing a raw data or copying the data including photo copy is strictly banned. Therefore, we could not upload raw data in Plos One submission system. I added dataset as supporting information, but raw data could not be included. Any research who had interest in this study can request the use of NHIS-NSC by following the procedures outlined at homepage of HIRA (https://opendata.hira.or.kr/op/opc/ selectPatDataAplInfoView.do). There is a cost for the use of virtual terminal.
(XLSX)

## Acknowledgments

The authors appreciate the statistical advice provided by the Medical Research Collaborating Center at Seoul National University Hospital.

## Author Contributions

**Conceptualization:** Chi Heon Kim, Chun Kee Chung, Yunhee Choi, Seung Heon Yang, Chang Hyun Lee, Sung Bae Park, Kyoung-Tae Kim, John M. Rhee, Moon Soo Park.

**Data curation:** Chi Heon Kim, Yunhee Choi, Juhee Lee, Seung Heon Yang, Chang Hyun Lee, Sung Bae Park, Kyoung-Tae Kim, John M. Rhee, Moon Soo Park.

**Formal analysis:** Chi Heon Kim, Chun Kee Chung, Yunhee Choi.

**Funding acquisition:** Chi Heon Kim, Chun Kee Chung.

**Investigation:** Chi Heon Kim.

**Methodology:** Chi Heon Kim, Chun Kee Chung.

**Project administration:** Chun Kee Chung.

**Software:** Juhee Lee.

**Supervision:** Chun Kee Chung.

**Validation:** Chun Kee Chung.

**Visualization:** Chun Kee Chung.

**Writing – original draft:** Chi Heon Kim, Chun Kee Chung, Yunhee Choi, Seung Heon Yang, Chang Hyun Lee, Sung Bae Park, Kyoung-Tae Kim, John M. Rhee, Moon Soo Park.

**Writing – review & editing:** Chi Heon Kim, Chun Kee Chung, Sung Bae Park, Kyoung-Tae Kim, Moon Soo Park.

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
