## [Decision Letter · Decision Letter 0]

7 Jul 2021

PONE-D-21-19512

Direct medical costs after surgical or nonsurgical treatment for degenerative lumbar spinal disease: a nationwide matched cohort study with a 10-year follow-up

PLOS ONE

Dear Dr. Chung,

Thank you for submitting your manuscript to PLOS ONE. After careful consideration, we feel that it has merit but does not fully meet PLOS ONE’s publication criteria as it currently stands. Therefore, we invite you to submit a revised version of the manuscript that addresses the points raised during the review process.

I would be happy to entertain a revision if you can adequately address the reviewers' comments.

We look forward to receiving your revised manuscript.

Kind regards,

Dean Chou, MD

Academic Editor

PLOS ONE

Journal Requirements:

2. In ethics statement in the manuscript and in the online submission form, please provide additional information about the patient records/samples used in your retrospective study. Specifically, please ensure that you have discussed whether all data/samples were fully anonymized before you accessed them and/or whether the IRB or ethics committee waived the requirement for informed consent. If patients provided informed written consent to have data/samples from their medical records used in research, please include this information.

Reviewers' comments:

Reviewer's Responses to Questions

**Comments to the Author**

1. Is the manuscript technically sound, and do the data support the conclusions?

Reviewer #1: Partly

Reviewer #2: Yes

2. Has the statistical analysis been performed appropriately and rigorously? 

Reviewer #1: Yes

Reviewer #2: Yes

3. Have the authors made all data underlying the findings in their manuscript fully available?

Reviewer #1: No

Reviewer #2: No

4. Is the manuscript presented in an intelligible fashion and written in standard English?

Reviewer #1: Yes

Reviewer #2: Yes

5. Review Comments to the Author

Reviewer #1: This study reviews the cost of surgical versus non-surgical intervention on lumbar degenerative spine pathology in a national database. The authors utilize Korea’s national health insurance data repository to evaluate fee for service charges. The authors find that, while surgery is initially more expensive, the results even out over years.

The manuscript offers good analysis of a national insurance database. It is certainly novel, and the results likely warrant publication. However, there are some issues.

When matching patients, the authors use age, sex, diagnosis and the presence of osteoporosis and diabetes as variables. The authors don’t give a rationale for these comorbidities. They leave out several important comorbidities (for example, smoking status, hypertension, CHF, etc). Furthermore, while it may not be available in their database, some sort of socioeconomic matching variable would be beneficial (insurance carrier, income, employment status, etc). At the very least, further explanation of their process for arriving at the matching variables is needed.

When matching the patients, are the costs over the last three years just regarding their spinal pathology or all medical costs?

The description of the “non-surgery” cohort as the “intervention” cohort is confusing. Why did the authors choose this label instead of something like “nonsurgical” or “conservative management?”

The authors do mention this, but it is worth reiterating: the costs here do not account for societal costs such as missed work and quality of life.

Lastly, the authors use charges reported to a national insurance database as a surrogate for costs here. This is a major point and warrants significant discussion. Charges are not a surrogate for costs, at least in the American system to which I am familiar. I highly doubt they are in Korea, either. The cost of a procedure or service indicates how much the provider must spend to provide that service, such as infrastructure, salaries, medications, and supplies.

Readers of this manuscript are likely going to be unfamiliar with the Korean medical financing system, so the authors should briefly explain this. If the hospitals are private entities, what constraints do they have on their charges? How do they determine the charge for a procedure or visit? How closely do the authors believe the charges represent the true cost of providing a procedure or visit? If they are freely set by the private institutions and responsive to market pressures, they are likely a good approximation. If they are set by the government or some planning body, they are unlikely to actually represent costs. The manuscript would benefit greatly from the authors both giving a brief explanation of the Korean healthcare financing system and of discussing how this would affect the relationship of charges to costs for this study.

Reviewer #2: This is an interesting study especially with long follow-up up to 10 years. Several issues that need to be addressed include the following:

- The cost of non-surgical management varies across countries and that contributes significantly to the data presentation.

- The extent of patients identified as disabled who did not return to work needs to be analyzed because the indirect cost of a leave of absence can offset some of the early costs of surgery.

- The reoperation rate was especially high in the surgical group, necessitating better clarification/stratification of this data for interpretation.

- Figure 2 and tables are unclear and needs better organization and clarification.

- State the percentage of patients who had long term follow.

- Determine the number of patients at each follow-up point.

- The quality of life and disability index score such as ODI would be helpful to better compare both groups.

6. PLOS authors have the option to publish the peer review history of their article (what does this mean?). If published, this will include your full peer review and any attached files.

Reviewer #1: **Yes: **Anthony DiGiorgio

Reviewer #2: No

---

## [Author Response · Author response to Decision Letter 0]

17 Sep 2021

Thank you very much for giving a chance to revise manuscript. We respected comments from the reviewers and modified manuscript accordingly. I added dataset as supporting information in revision, but raw data could not be included. The National Health Insurance System-National Sample Cohort (NHIS-NSC) was utilized for this study after approval by health insurance review and assessment service (HIRA). Individual data linkage of population was made internally in the Big Data Steering Department of the National Health Insurance Service. The authors of the study were approved to use customized tables via virtual terminal connected to personal computer after review of study proposal by HIRA for less than 6 months. By law, sharing a raw data or copying the data including photo copy is strictly banned. Therefore, we could not upload raw data in Plos One submission system. Any research who had interest in this study can request the use of NHIS-NSC by following the procedures outlined at homepage of HIRA (https://opendata.hira.or.kr/op/opc/selectPatDataAplInfoView.do). The responds to each point raised by the academic editor and reviewer were answered in a separate file labeled 'Response to Reviewers'.

---

## [Decision Letter · Decision Letter 1]

21 Oct 2021

PONE-D-21-19512R1Direct medical costs after surgical or nonsurgical treatment for degenerative lumbar spinal disease: a nationwide matched cohort study with a 10-year follow-upPLOS ONE

Dear Dr. Chung,

Thank you for submitting your manuscript to PLOS ONE. After careful consideration, we feel that it has merit but does not fully meet PLOS ONE’s publication criteria as it currently stands. Therefore, we invite you to submit a revised version of the manuscript that addresses the points raised during the review process.

Please address reviewer #1's comments.

We look forward to receiving your revised manuscript.

Kind regards,

Dean Chou, MD

Academic Editor

PLOS ONE

Journal Requirements:

Reviewers' comments:

Reviewer's Responses to Questions

**Comments to the Author**

1. If the authors have adequately addressed your comments raised in a previous round of review and you feel that this manuscript is now acceptable for publication, you may indicate that here to bypass the “Comments to the Author” section, enter your conflict of interest statement in the “Confidential to Editor” section, and submit your "Accept" recommendation.

Reviewer #1: (No Response)

Reviewer #2: All comments have been addressed

2. Is the manuscript technically sound, and do the data support the conclusions?

Reviewer #1: Partly

Reviewer #2: Yes

3. Has the statistical analysis been performed appropriately and rigorously? 

Reviewer #1: Yes

Reviewer #2: Yes

4. Have the authors made all data underlying the findings in their manuscript fully available?

Reviewer #1: Yes

Reviewer #2: No

5. Is the manuscript presented in an intelligible fashion and written in standard English?

Reviewer #1: Yes

Reviewer #2: Yes

6. Review Comments to the Author

Reviewer #1: The authors have done an outstanding job addressing most of my previous concerns. My one lasting issue is with the discussion of “cost” versus “charges.” The authors have included an excellent overview of the Korean healthcare financing system. I applaud them for that. From the author’s explanation, the costs they report are based on a centrally set reimbursement fee by the Korean government. Thus, they don’t represent true costs. They state in their response to my comment that the “cost” of medical services and supplies is determined by the board of the NHIS. A more accurate description would be that the “reimbursement” is determined by the board of NHIS. This is similar to the United States. The government may set a reimbursement for a surgical procedure, but that reimbursement is not reflective of the true cost. The administrators who set that rate may try to approximate cost, but costs are ever shifting and determined by the market. If, for example, the cost of labor increases (as has happened recently), the costs of surgical procedures will increase. However, the reimbursement rate from the government will remain the same, regardless of how much the hospital must pay out for staffing, electricity, supplies, etc. True determination of cost requires a thorough accounting including time-driven activity-based costing, which is far outside the scope of this paper.

While these economic nuances need not be included in this particular manuscript, we should ensure that the conversation around surgical costs is appropriately phrased. For example, in the conclusion, instead of stating “surgery was initially more costly,” it would be more appropriate to say “surgical treatment initially led to more government reimbursement than nonsurgical…” or “cost the government payor more…” or something along those lines. The authors should ensure that the language in the paper does not suggest that the monetary figures are implied to be actual economic costs but instead reflect reimbursement charges from the government.

Reviewer #2: (No Response)

7. PLOS authors have the option to publish the peer review history of their article (what does this mean?). If published, this will include your full peer review and any attached files.

Reviewer #1: **Yes: **Anthony M DiGiorgio, DO, MHA

Reviewer #2: No

---

## [Author Response · Author response to Decision Letter 1]

26 Oct 2021

Reviewer #1: The authors have done an outstanding job addressing most of my previous concerns. My one lasting issue is with the discussion of “cost” versus “charges.” The authors have included an excellent overview of the Korean healthcare financing system. I applaud them for that. From the author’s explanation, the costs they report are based on a centrally set reimbursement fee by the Korean government. Thus, they don’t represent true costs. They state in their response to my comment that the “cost” of medical services and supplies is determined by the board of the NHIS. A more accurate description would be that the “reimbursement” is determined by the board of NHIS. This is similar to the United States. The government may set a reimbursement for a surgical procedure, but that reimbursement is not reflective of the true cost. The administrators who set that rate may try to approximate cost, but costs are ever shifting and determined by the market. If, for example, the cost of labor increases (as has happened recently), the costs of surgical procedures will increase. However, the reimbursement rate from the government will remain the same, regardless of how much the hospital must pay out for staffing, electricity, supplies, etc. True determination of cost requires a thorough accounting including time-driven activity-based costing, which is far outside the scope of this paper.

While these economic nuances need not be included in this particular manuscript, we should ensure that the conversation around surgical costs is appropriately phrased. For example, in the conclusion, instead of stating “surgery was initially more costly,” it would be more appropriate to say “surgical treatment initially led to more government reimbursement than nonsurgical…” or “cost the government payor more…” or something along those lines. The authors should ensure that the language in the paper does not suggest that the monetary figures are implied to be actual economic costs but instead reflect reimbursement charges from the government.

Answer. 

I appreciate for the reviewer’s concern and valuable suggestion. I totally agree with your opinion that the “reimbursement” is determined by the board of NHIS. Hence, this paper has a limitation that true determination of cost requires a thorough accounting including time-driven activity-based costing, which is far outside the scope of this paper. Therefore, I clarified that the costs in this paper means a charge to government for a reimbursement, which does not reflect the actual economic cost in the revised manuscript including limitation section. 

Abstract 

Results

The characteristics and matching factors were well-balanced between the matched cohorts. Overall, surgery cohort spent $50.84/patient/month, while the nonsurgical cohort spent $29.34/patient/month (p<0.01). Initially, surgery treatment led to more charge to NHIS ($2,762) than nonsurgical treatment ($180.4) (p<0.01). Compared with the non-surgical cohort, the surgery cohort charged $33/month more for the first 3 months, charged less at 12 months, and charged approximately the same over the course of 10 years. 

Conclusion

Surgical treatment initially led to more government reimbursement than nonsurgical treatment, but the charges during follow-up period were not different. The results of the present study should be interpreted in light of the costs of medical services, indirect costs, societal cost, quality of life and societal willingness to pay in each country. The monetary figures are implied to be actual economic costs but those in the reimbursement system instead reflect reimbursement charges from the government. 

Page 9, line 8

During the last 3 years, the standard deviation of the direct medical cost (charge to NHIS) in 3,881 patients was $730, and a difference of no more than $73 was considered a similar severity of disease in the matching process.

Page 11, line 22

Therefore, the imbursement by NHIS does not represent actual cost of hospital but represents a utilization of NHIS.

Page 22, line 5

Lastly, but not least, this paper has a limitation that true determination of cost requires a thorough accounting including time-driven activity-based costing, which is far outside the scope of this paper. The present monetary figures were a charge to government for a reimbursement, which does not reflect the actual economic cost. 

Conclusion 

Surgical treatment initially led to more government reimbursement than nonsurgical treatment, but the charges during follow-up period were not different. The results of the present study should be interpreted in light of the costs of medical services, indirect costs, societal cost, quality of life and societal WTP in each country. The monetary figures are implied to be actual economic costs but those in the reimbursement system instead reflect reimbursement charges from the government.

Reviewer #2: (No Response)

Answer. 

I appreciate your advices for the manuscript.

---

## [Decision Letter · Decision Letter 2]

10 Nov 2021

Direct medical costs after surgical or nonsurgical treatment for degenerative lumbar spinal disease: a nationwide matched cohort study with a 10-year follow-up

PONE-D-21-19512R2

Dear Dr. Chung,

Congratulations!  We’re pleased to inform you that your manuscript has been judged scientifically suitable for publication and will be formally accepted for publication once it meets all outstanding technical requirements.

Kind regards,

Dean Chou, MD

Academic Editor

PLOS ONE

Additional Editor Comments (optional):

Reviewers' comments:

Reviewer's Responses to Questions

**Comments to the Author**

1. If the authors have adequately addressed your comments raised in a previous round of review and you feel that this manuscript is now acceptable for publication, you may indicate that here to bypass the “Comments to the Author” section, enter your conflict of interest statement in the “Confidential to Editor” section, and submit your "Accept" recommendation.

Reviewer #1: All comments have been addressed

2. Is the manuscript technically sound, and do the data support the conclusions?

Reviewer #1: Yes

3. Has the statistical analysis been performed appropriately and rigorously? 

Reviewer #1: Yes

4. Have the authors made all data underlying the findings in their manuscript fully available?

Reviewer #1: Yes

5. Is the manuscript presented in an intelligible fashion and written in standard English?

Reviewer #1: Yes

6. Review Comments to the Author

Reviewer #1: The authors have done an outstanding job updating the manuscript. I have no further comments.

7. PLOS authors have the option to publish the peer review history of their article (what does this mean?). If published, this will include your full peer review and any attached files.

Reviewer #1: **Yes: **Anthony DiGiorgio

---

## [Editor Report · Acceptance letter]

18 Nov 2021

PONE-D-21-19512R2 

Direct medical costs after surgical or nonsurgical treatment for degenerative lumbar spinal disease: a nationwide matched cohort study with a 10-year follow-up 

Dear Dr. Chung:

I'm pleased to inform you that your manuscript has been deemed suitable for publication in PLOS ONE. Congratulations! Your manuscript is now with our production department. 

Kind regards, 

on behalf of

Dr. Dean Chou 

Academic Editor

PLOS ONE